# Modelling and Multi-Objective Optimization of the Sulphur Dioxide Oxidation Process

**Mohammad Reza Zaker, Clémence Fauteux-Lefebvre and Jules Thibault \***

Department of Chemical and Biological Engineering, University of Ottawa, Ottawa, ON K1N 6N5, Canada; mzake089@uottawa.ca (M.R.Z.); cfauteux@uottawa.ca (C.F.-L.)

**\*** Correspondence: jules.thibault@uottawa.ca

**Abstract:** Sulphuric acid ($H_2SO_4$) is one of the most produced chemicals in the world. The critical step of the sulphuric acid production is the oxidation of sulphur dioxide ($SO_2$) to sulphur trioxide ($SO_3$) which takes place in a multi catalytic bed reactor. In this study, a representative kinetic rate equation was rigorously selected to develop a mathematical model to perform the multi-objective optimization (MOO) of the reactor. The objectives of the MOO were the $SO_2$ conversion, $SO_3$ productivity, and catalyst weight, whereas the decisions variables were the inlet temperature and the length of each catalytic bed. MOO studies were performed for various design scenarios involving a variable number of catalytic beds and different reactor configurations. The MOO process was mainly comprised of two steps: (1) the determination of Pareto domain via the determination a large number of non-dominated solutions, and (2) the ranking of the Pareto-optimal solutions based on preferences of a decision maker. Results show that a reactor comprised of four catalytic beds with an intermediate absorption column provides higher $SO_2$ conversion, marginally superior to four catalytic beds without an intermediate $SO_3$ absorption column. Both scenarios are close to the ideal optimum, where the reactor temperature would be adjusted to always be at the maximum reaction rate. Results clearly highlight the compromise existing between conversion, productivity and catalyst weight.

**Keywords:** packed bed reactor; multi-objective optimization; $SO_2$ kinetic rate equations; non dominated solutions; pareto domain; $SO_2$ oxidation process

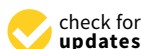



## 1. Introduction

Sulphuric acid ($H_2SO_4$) is a chemical compound of paramount industrial importance and used in the fabrication of a myriad of products. Sulphuric acid is a strong mineral acid. It is a colorless and viscous liquid that is soluble in water at all concentrations [1]. Sulphuric acid is a very important commodity chemical and a nation's sulphuric acid production is one of the indicators of its industrial strength [2]. To minimize the risk associated with its transportation, sulphuric acid production plants are usually located near their point of use, as it is a corrosive and risky chemical to transport. Indeed, it is more economical and safer to transport elemental sulphur than sulphuric acid [2]. Sulphuric acid is an essential chemical for numerous process industries, such as fertilizer manufacturing, oil refining, chemical synthesis, pharmaceuticals, and lead-acid batteries [1].

Originally, sulphuric acid was produced using the "Chamber" process [3] where the oxidation of sulphur dioxide ($SO_2$) with moist air was oxidized using nitrogen oxides as the catalyst [2]. The reaction took place in a series of large, lead-lined chambers [2]. Due to the small plant size and product acid strength limited to 70% sulphuric acid [4], it has been largely supplanted in modern industrial processes by the "Contact" process [5]. The simplified flowsheet of the current industrial sulphuric acid production is shown in Figure 1. The $H_2SO_4$ manufacturing is comprised of three main steps. In the first step, elemental sulphur is burned in a furnace in the presence of dry air to produce $SO_2$. In the second step, $SO_2$ is oxidized to $SO_3$ in an adiabatic packed bed reactor using vanadium

pentoxide catalyst. In the final step, an absorption column of concentrated $H_2SO_4$ is used, where $SO_3$ reacts with water to form $H_2SO_4$. The chemical reactions associated with these steps are given in Equations (1)–(3) along with their respective heat of reaction [1]. It is important to note that the three reactions are highly exothermic, with the second reaction being an equilibrium reaction. The conversion of $SO_3$ to $H_2SO_4$ is performed using a very concentrated sulphuric acid solution, and not directly with water, due to the highly exothermic nature of the reaction. In this third reaction, oleum ($H_2S_2O_7$) will also be formed, but upon reacting with water leads to $H_2SO_4$. Most plants and smaller units of $H_2SO_4$ production use a single final absorption process, which leads to a conversion of 96–98%. Some modern large-capacity sulphuric acid production plants utilize a double contact absorption process [6]. By adding an intermediate absorption column, the double contact process can achieve a conversion in excess of 99%, but at the expense of a more complex process [2].

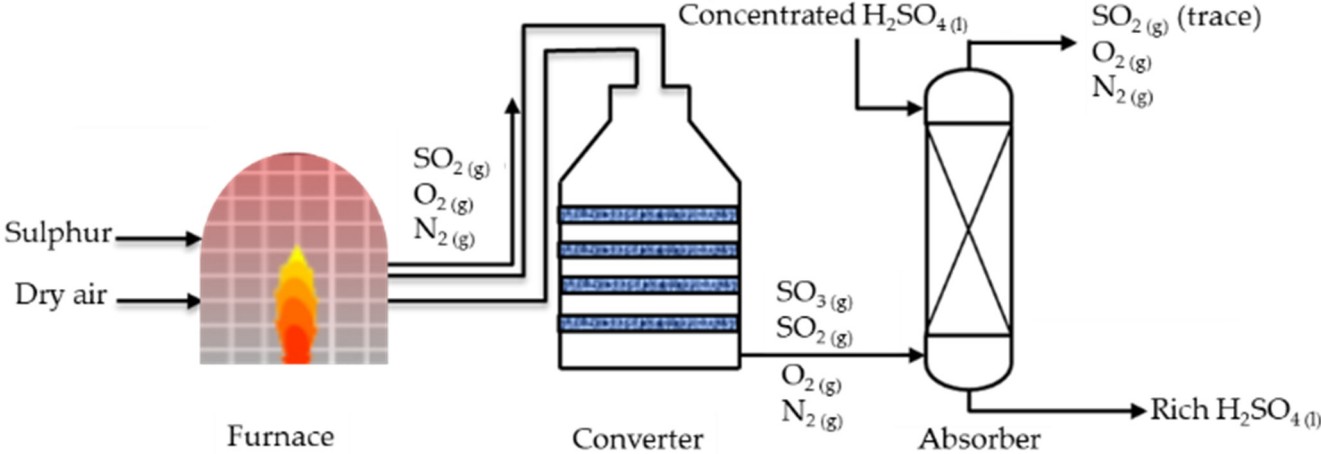

**Figure 1.** Simplified sulphuric acid production flowsheet showing the main three steps: (1) combustion of sulphur to produce $SO_2$ in presence of dry air, (2) oxidation of $SO_2$ to $SO_3$ in a series of adiabatic catalytic packed beds, and (3) absorption of sulphur trioxide in a concentrated sulphuric acid absorber and conversion to sulphuric acid.

The critical step in the production of $H_2SO_4$ is the oxidation of $SO_2$ to $SO_3$. As this reaction is a highly exothermic equilibrium reaction, it takes place in a series of sequential adiabatic catalytic beds with external intercoolers to decrease the gas mixture temperature prior to entering the next catalytic bed in order to increase the $SO_2$ oxidation rate. Platinum was initially the most used catalyst for this reaction [7]. However, the cost of platinum and its susceptibility to poisoning by trace elements led to the development of the vanadium pentoxide catalyst, which is currently widely used in industrial processes [8].

$$S_{(s)} + O_{2\ (g)} \rightarrow SO_{2\ (g)} \ \Delta H_R = -296{,}810 \ Kj/kmol \tag{1}$$

$$SO_{2\ (g)} + 0.5O_{2\ (g)} \rightleftharpoons SO_{3\ (g)} \ \Delta H_R = -96{,}232 \ Kj/kmol \tag{2}$$

$$SO_{3\ (g)} + H_2O_{(l)} \rightarrow H_2SO_{4\ (l)} \ \Delta H_R = -132{,}000 \ Kj/kmol \tag{3}$$

The typical progression of the $SO_2$ conversion and the bed temperature for this exothermic equilibrium reaction in a four-bed adiabatic catalytic plug flow reactor is presented in Figure 2. The location of the equilibrium curve in Figure 2 depends on the inlet composition and pressure of the reacting mixture. The slope of the adiabatic operating lines is a function of the thermal capacity of the reacting mixture and the heat of reaction [9]. The horizontal operating lines represent the cooling of the reaction mixture, which takes place between two sequential beds in external heat exchangers [10]. The curve of the maximum rate is the locus of the conversion-temperature coordinates at which the reaction rate is maximum [10].

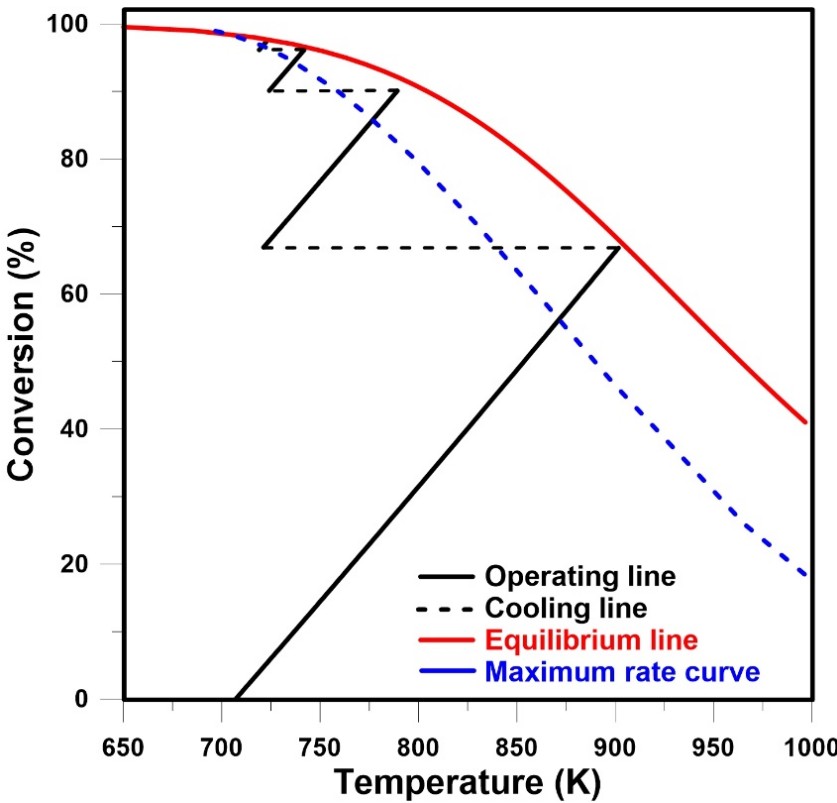

**Figure 2.** Typical conversion versus temperature diagram of a four catalytic bed process used to conduct the $SO_2$ to $SO_3$ reaction with gas cooling between catalytic beds. The equilibrium and maximum rate curves are shown in red and blue, respectively.

Given the complexity of the $SO_2$ to $SO_3$ reactor configuration, it is paramount to adopt the optimal configuration and to operate the process under optimal conditions. It is obviously desired to maximize the $SO_2$ conversion, as unreacted $SO_2$ will need to be treated or recover at the exit of the $SO_3$ absorption column. However, a high conversion rate can only be achieved at the expense of a lower productivity and a larger reactor volume, leading to a set of competitive objectives; productivity needs to be maximized, whereas the reactor volume needs to be minimized. Therefore, to resolve these conflicting objectives, a multi-objective optimization (MOO) can be used to strike a suitable compromise between the various objectives that would be satisfactory in the eye of a decision-maker. MOO allows for the optimization of multiple and often conflicting objectives, which produces a set of alternative solutions, called the Pareto domain. These solutions, obtained without any bias as to the importance of each objective, are said to be Pareto-optimal, in the sense that no one solution is better than any other in the domain when compared on all objectives [11]. The experience and knowledge of the decision-maker are then used to rank the entire Pareto domain. The application of MOO techniques in chemical engineering have demonstrated their strength in the prediction of optimum operating conditions in the presence of multiple conflicting objectives [12]. For example, Caceres et al. [13] and Vandervoort et al. [14] successfully employed MOO to optimize acrylic acid production and an ethylene oxide reactor, respectively. This work, therefore, resorts to a MOO algorithm to investigate the impact of the number of catalytic packed beds and the addition of an intermediate absorption column for conducting this oxidation reaction. The MOO algorithm allows the comparison of the optimal solution of each studied strategy based on various process objectives. In the case of the $SO_2$ to $SO_3$ multi-bed catalytic reactor, the decision variables could be the inlet temperature and the length of each catalytic bed, the reactor pressure, and the configuration of the packed catalytic beds.

To properly study, control, and optimize the multi-bed oxidation catalytic reactor for the $SO_2$ to $SO_3$ reaction, it is paramount to identify a representative and reliable kinetic rate equation that is able to accurately predict the fate of reacting molecules for the range of operating conditions encountered in an industrial reactor. Such models are useful for the optimization of the process by identifying the optimal operating conditions which lead to the best process performance metrics. For the $SO_2$ oxidation on vanadium pentoxide ($V_2O_5$) catalyst, a number of kinetic models have been proposed, such as the ones from Collina et al. [15], Eklund [16], Calderbank [17], and Villadsen et al. [18], each obtained over a limited range of operating conditions. Various model evaluation techniques can be used to find the appropriateness of each model. The minimization of the residual sum of squares (RSS) is used in this study to objectively evaluate the discrepancy between the experimental data and a given kinetic model, and select the model that best predicts the experimental data over the operating range of interest.

This investigation aims to perform the multi-objective optimization of the $SO_2$ to $SO_3$ multi-bed catalytic reactor. The first step in the optimization process is to select the most representative dynamic model of the industrial catalytic bed reactor and rate equation. Secondly, a multi-objective optimization for a set of decision variables and objectives is conducted to circumscribe and rank the Pareto domain for a series of process strategies. Finally, the performance of the various process strategies is presented and compared.

## 2. Kinetics of Reaction

The first step to perform the multi-objective optimization of the multi-bed catalytic reactor was selecting a kinetic model that is representative of the $SO_2$ to $SO_3$ reaction under industrial operating conditions. Two approaches are normally used for this selection. The first is to use the intrinsic reaction kinetics, which is scale independent. With this first approach, it is required to consider the heat and mass transfer phenomena to calculate the temperature and concentration profiles within the catalyst particles. The second approach is to perform experiments using the industrial catalyst and operating conditions in a pilot-scale reactor or the actual reactor, for which the kinetic model already embeds the heat and mass transfer limitations. In this investigation, the second approach was favored as the kinetics models have mostly been derived for experiments conducted under industrially relevant conditions, which consider simultaneously the effect of heat and mass transfer, the geometry of the catalyst particles, and the intrinsic kinetics of reaction.

A literature survey has led to four kinetic models, which were derived based on experience performed with the industrial vanadium pentoxide catalyst particles under industrially relevant operating conditions. The kinetic models published by Collina et al. [15], Eklund [16], Calderbank [17], and Villadsen et al. [18] were identified to be potentially representative kinetic rate models for the present investigation. The equations of these four kinetic rate models are given in Equations (4)–(7), respectively. The parameters of the kinetic rate expressions are given in Table 1. To investigate the appropriateness of the kinetic rate models, and assist in the selection of the most representative reaction rate kinetic model for the $SO_2$ oxidation, the experimental data published by Doering et al. [19] were used. A total of 135 experimental data points, obtained with the industrial vanadium pentoxide catalyst, were available to test the models. The experimental data of Doering et al. [19] provided the rate of reaction for conversions above 97%, pressures up to 10 atm, and temperatures in the range of 360–400 °C.

$$-r_{SO_2} = \frac{k_1 \cdot p_{O_2} \cdot p_{SO_2} \left(1 - \frac{p_{SO_3}}{K_P \cdot p_{SO_2} \cdot p_{O_2}^{0.5}}\right)}{22.414(1 + K_2 \cdot p_{SO_2} + K_3 \cdot p_{SO_3})^2} \left(\frac{kmol}{kg_{cat} \cdot s}\right) \tag{4}$$

$$-r_{SO_2} = k_1 \left(\frac{p_{SO_2}}{p_{SO_3}}\right)^{0.5} \left[p_{O_2} - \left(\frac{p_{SO_3}}{K_P \cdot p_{SO_2}}\right)^2\right] \left(\frac{kmol}{kg_{cat} \cdot s}\right) \tag{5}$$

$$-r_{SO_2} = \frac{k_1 p_{O_2} p_{SO_2} - k_2 p_{SO_3} p_{O_2}^{0.5}}{p_{SO_2}^{0.5}} \left( \frac{kmol}{kg_{cat} \cdot s} \right) \tag{6}$$

$$-r_{SO_2} = \frac{k_1 \cdot p_{O_2}^{0.5} \cdot \left( 1 - \frac{p_{SO_3}}{K_p \cdot p_{SO_2} \cdot p_{O_2}^{0.5}} \right)}{K_2 + K_3 \cdot p_{SO_3} + \frac{p_{SO_3}}{p_{SO_2}}} \left( \frac{kmol}{kg_{cat} \cdot s} \right) \tag{7}$$

**Table 1.** Expressions of kinetic parameters and their temperature range of validity.

| Parameters | Rate Constant Expressions | T (°C) |
|---|---|---|
| Model 1: Collina et al. [15] | | |
| $k_1 \left[ \frac{kmol}{kg_{cat} \cdot atm^2 \cdot s} \right]$ | $k_1 = \exp(12.16 - \frac{5473}{T})$ | |
| $k_2 \left[ atm^{-1} \right]$ | $k_2 = \exp(-9.953 + \frac{8619}{T})$ | |
| $k_3 \left[ atm^{-1} \right]$ | $k_3 = \exp(-71.745 + \frac{52596}{T})$ | 420–590 |
| $k_P \left[ atm^{-0.5} \right]$ | $k_P = \exp(-10.68 + \frac{11300}{T})$ | |
| Model 2: Eklund [16] | | |
| $k_1 \left[ \frac{kmol}{kg_{cat} \cdot atm \cdot s} \right]$ | $k_1 = \exp(848.14 - \frac{97782.2}{T} - 110.1 \ln T)$ | 420–554 |
| $k_P \left[ atm^{-0.5} \right]$ | $k_p = \exp(-11.24 + \frac{11818.055}{T})$ | |
| Model 3: Calderbank [17] | | |
| $k_1 \left[ \frac{kmol}{kg_{cat} \cdot atm^{1.5} \cdot s} \right]$ | $k_1 = \exp(12.07 - \frac{15656.56}{T})$ | 370–450 |
| $k_2 \left[ \frac{kmol}{kg_{cat} \cdot atm \cdot s} \right]$ | $k_2 = \exp(-22.75 + \frac{27070.7}{T})$ | |
| Model 4: Villadsen et al. [18] | | |
| $k_1 \left[ \frac{kmol}{kg_{cat} \cdot atm \cdot s} \right]$ | $k_1 = \exp(-1.88 - \frac{7466.08}{T})$ | |
| $k_2 [-]$ | $k_2 = \exp(2.10 + \frac{286.74}{T})$ | 380–520 |
| $k_3 \left[ atm^{-1} \right]$ | $k_3 = \exp(-1.51 + \frac{2279.17}{T})$ | |
| $K_P \left[ atm^{-0.5} \right]$ | $K_p = \exp(-10.73 + \frac{11318.3}{T})$ | |

The four kinetic rate models were compared with experimental data obtained at 400 °C for a feed composition of 11 mol% $SO_2$ and 10 mol% $O_2$ at a pressure of 2.5 and 10 atm, as well for a gas feed composition of 10 mol% $SO_2$ and 11 mol% $O_2$ at 1.12 atm, representing the range of operating conditions under study [19]. Results of this comparison are presented in Figure 3. Two important observations can be made from these results. First, the Eklund kinetic model better represents the experimental data at a pressure near 10 atm (Figure 3a). However, the kinetic model of Collina et al. predicts with better accuracy the experimental data at the two lower pressures, including near atmospheric pressure (Figure 3b,c).

Similar results were obtained at other experimental conditions. One can conclude that there is no unique reaction kinetic model that represents the kinetic data over the entire range of the experimental conditions. To better assess the representativeness of the different reaction kinetic models for the complete experimental data set, the residual sum of squares of the relative differences between the predicted and experimental data was calculated (Equation (8)) and used to assess the predictive performance of each kinetic model.

$$RSS = \sum_{i=1}^{N} \left( \frac{r_{e,SO_{2_i}} - r_{SO_{2_i}}}{r_{e,SO_{2_i}}} \right)^2 \tag{8}$$

where $r_{e,SO_{2_i}}$ is the experimental rate of i-th experimental point, $r_{SO_{2_i}}$ is the calculated rate of the i-th experimental point according to the reaction mechanism model, and N is the number of experimental points considered. The residual sum squares of the relative differences for each kinetic model for different operating conditions are presented in Figure 4. Figure 4a shows the RSS for different temperatures at a constant pressure of 2.5 atm and a feed gas composition of 10 mol% $SO_2$ and 11 mol% $O_2$. When the temperature

is increased, the RSS decreases significantly for all kinetic models, which indicates that the models are more accurate if the reaction occurs above 400 °C, which is the case for the industrial reactor. The models of Collina et al. and Eklund have the lowest RSS values compared to other models. Figure 4b shows the performance of the four kinetic models when the reaction is conducted at different pressures. The model proposed by Collina et al. has the lowest RSS values at low operating pressures. When the pressure increases, the RSS values increase. In contrast, the models of Eklund and Calderbank show better predictions at higher pressures. By considering the entire experimental pressure range (1 to 10 atm) and the temperature range investigated, the Collina et al. kinetic model has a lower average RSS value and, in general, shows a better fit considering all the experimental data in comparison with other kinetic models (Table 2).

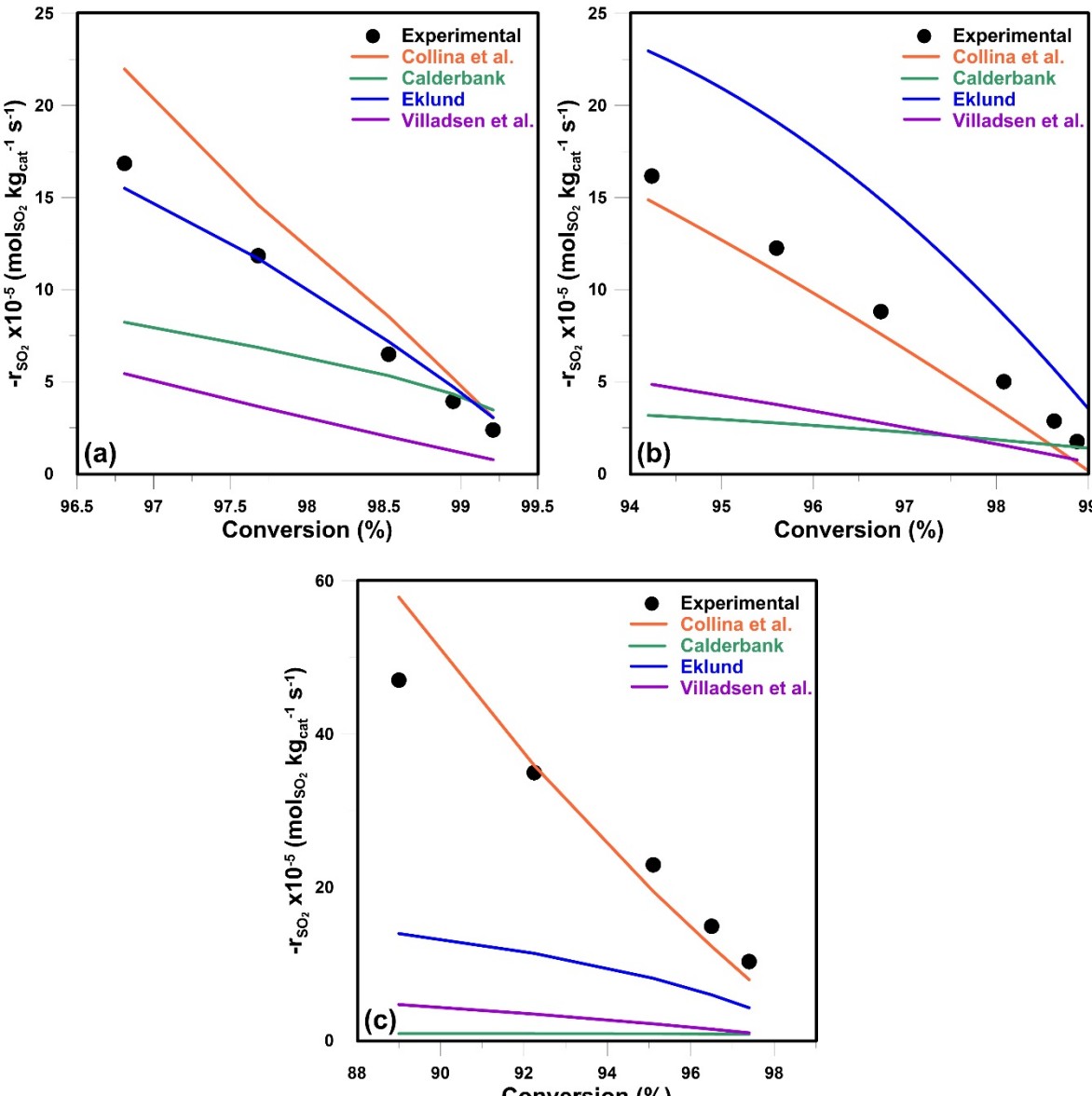

**Figure 3.** Comparison of the experimental and predicted reaction rates as a function of conversion at T = 400 °C: (**a**) 11 mol% $SO_2$, 10 mol% $O_2$, and P = 10 atm; (**b**) 11 mol% $SO_2$, 10 mol% $O_2$, and P = 2.5 atm; and (**c**) 10 mol% $SO_2$, 11 mol% $O_2$, and P = 1.12 atm.

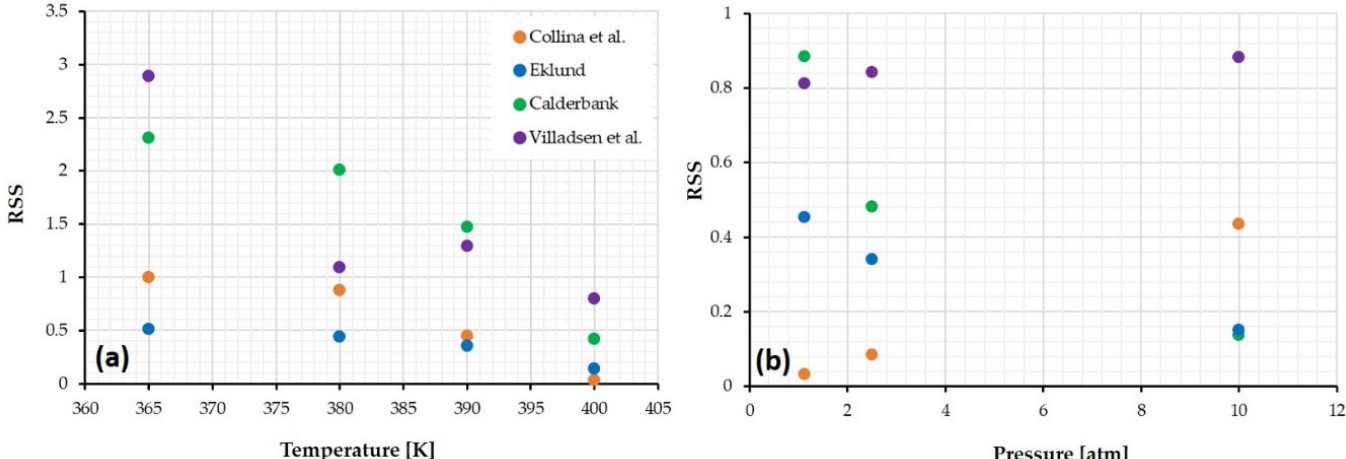

**Figure 4.** Residual sum squares (RSS) of each kinetic model for feed gas compositions of 10 mol% $SO_2$, 11 mol% $O_2$ for (**a**) different temperatures at a constant pressure of 2.5 atm, and (**b**) different pressures at the constant temperature (400 °C).

**Table 2.** Residual sum of squares for the four evaluated kinetic models over the entire range of the experimental data.

| Model Equation | Residual Sum of Squares (RSS) |
| --- | --- |
| Model 1: Collina et al. [15] | 0.756 |
| Model 2: Eklund [16] | 0.936 |
| Model 3: Calderbank [17] | 3.55 |
| Model 4: Villadsen et al. [18] | 2.08 |

Based on this analysis, this kinetic model of Collina et al. better represents the entire experimental data than the other models. To perform a multi-objective optimization (MOO) for $SO_2$ oxidation to $SO_3$ in the plug flow reactor, the model of Collina et al. was used in this investigation, especially for pressure near atmospheric levels and for the feed gas composition of 10 mol% $SO_2$, 11 mol% $O_2$.

### 3. Reactor Modeling

Industrially, the $SO_2$ to $SO_3$ oxidation is typically conducted adiabatically in a series of large-diameter packed catalytic reactors with inter-reactor heat exchangers. Each packed bed reactor has a height between 0.02 and 2.0 m, and a diameter varying between 5 and 12 m. It is assumed that ideal plug flow conditions prevail with negligible axial and radial dispersion, such that the integration of the mass and heat transfer equations is performed only in the axial direction. In addition, it is assumed that the reactor operates under steady state. The general molar balance equation describing the $SO_2$ conversion (X) along the axial direction of reactor bed as a function of the reaction rate, bulk density, the bed area, and the $SO_2$ molar flow rate, is given by Equation (9).

$$\frac{dX}{dz} = \frac{-r_{SO_2} \cdot \rho_b \cdot A}{\left(F_{SO_2}\right)_{in}} \tag{9}$$

As the $SO_2$ oxidation reaction is very exothermic and the reaction rate is strongly dependent on temperature, the steady-state one-dimensional energy balance must also be solved to determine the variation of temperature along the bed, as described in Equation (10). It is assumed that both the axial thermal dispersion and the radial gradient are negligible.

$$\frac{dT}{dz} = \frac{-r_{SO_2} \cdot \rho_b \cdot A \cdot \Delta H_R}{C_{P\,Mii}} \tag{10}$$

The rate of reaction depends on the partial pressure of the two reactants and the product. To determine the pressure drop through the packed beds, the Ergun equation [20], giving the change in pressure as a function of the bed length, is used (Equation (11)).

$$\frac{dP}{dz} = 150 \times \frac{\mu \cdot u \cdot (1 - \varepsilon)^2}{D_P^2 \cdot \varepsilon^3} + 1.75 \times \frac{\rho_f \cdot u^2 \cdot (1 - \varepsilon)}{D_P \cdot \varepsilon^3} \tag{11}$$

where $C_{p,Mix}$, $\Delta H_R$, and $\mu$ are mixture heat capacity, heat of reaction and gas viscosity, respectively. Their temperature dependencies are discussed in Appendices A–C, respectively.

To determine the conversion, temperature, and pressure as a function of position along the packed bed, the packed bed is discretized in a number of thin slices. For each slice, the heat of reaction, heat capacity, viscosity, and reaction rate coefficients with their respective temperature dependencies are determined, and the operating conditions are therefore set to carry out the calculations for the next slice of the catalytic bed reactor. The mass and energy balance equations, and the Ergun equation is numerically integrated, starting with the inlet conditions of the gas mixture until the end of the last catalytic bed. The $SO_2$ conversion, $SO_3$ productivity, and catalyst weight are then evaluated as described in Equations (12)–(14), respectively.

$$\text{Conversion } (X) = \frac{\left(F_{SO_2}\right)_{in} - \left(F_{SO_2}\right)_{out}}{\left(F_{SO_2}\right)_{in}} \times 100 \ (\%) \tag{12}$$

$$\text{Productivity } (\text{Pro}) = \frac{F_{SO_3}}{A \times \rho_b \times \sum L_i} \left(\frac{mol}{kg_{cat} \cdot s}\right) \tag{13}$$

$$\text{Catalyst weight } (W) = A \times \rho_b \times \sum L_i \ (kg) \tag{14}$$

*Operating Conditions*

In this study, the pressure of the inlet gas mixture to the sulphur dioxide converter was set slightly higher than atmospheric pressure, namely at 1.4 atm, in the validity range of the selected kinetic model. A 12 m diameter adiabatic catalytic reactor was selected with a feed gas flow rate of 800 mol/s. The inlet temperature and the length of each catalytic bed are decision variables, and will be adjusted with the optimization algorithm in order to determine all Pareto-optimal solutions, each solution being comprised of the conversion, productivity, and catalyst weight. All simulation parameters, such as the bed porosity, bed bulk density, and gas mixture density, etc., were chosen according to the literature and are given in Table 3.

**Table 3.** Reactor system simulation parameters.

| Parameter | Description | Value |
|:---:|:---:|:---:|
| $\varepsilon$ | Bed porosity | 0.303 |
| $\rho_b$ | Bed density | 256 (kg/m$^3$) |
| $\rho_f$ (T = 460 °C) | Inlet gas density | 0.781 (kg/m$^3$) |
| $A$ | Reactor area | 113 (m$^2$) |
| $D_{bed}$ | Bed diameter | 12 (m) |
| $D_p$ | Equivalent particle diameter | $6.35 \times 10^{-3}$ (m) |
| $F_{in}$ | Inlet flow rate | 800 (mol/s) |
| $P_{in}$ | Inlet pressure | 1.4 (atm) |
| $R$ | Gas constant | 8.314 (Pa·m$^3$/mol·K) |
| $y_{N2,in}$ | Inlet mole fraction of N$_2$ | 0.79 |
| $y_{O2,in}$ | Inlet mole fraction of O$_2$ | 0.11 |
| $y_{SO2,in}$ | Inlet mole fraction of SO$_2$ | 0.1 |
| $y_{SO3,in}$ | Inlet mole fraction of SO$_3$ | 0 |

## 4. Multi-Objective Optimization Methodology

A multi-objective optimization approach is used to optimize the oxidation of $SO_2$ to $SO_3$. The optimization methodology comprises four main steps as illustrated in Figure 5 [21]. The optimization problem is first established by defining the set of objective functions to be maximized or minimized, and the set of decision variables with their respective allowable ranges. The next step is determining a representative process model to perform the optimization study. In this investigation, the process model is comprised of the integration along the axial direction of the steady-state heat and mass balances and the pressure drop equations (Equations (9)–(11)), which allows us to determine the three objective functions: conversion ($X$), productivity (Pro), and catalyst weight ($W$). Once the packed-bed catalytic reactor model is available, it is used to circumscribe the Pareto domain using a MOO algorithm. In this investigation, the Non-Sorting Genetic Algorithm II (NSGA II) has been used [22]. Finally, all Pareto-optimal solutions are ranked using the Net Flow Method (NFM) [11], where the preferences of a decision-maker are embedded in a set of relative weights and three threshold criteria of each objective to assist in the ranking of all Pareto-optimal solutions.

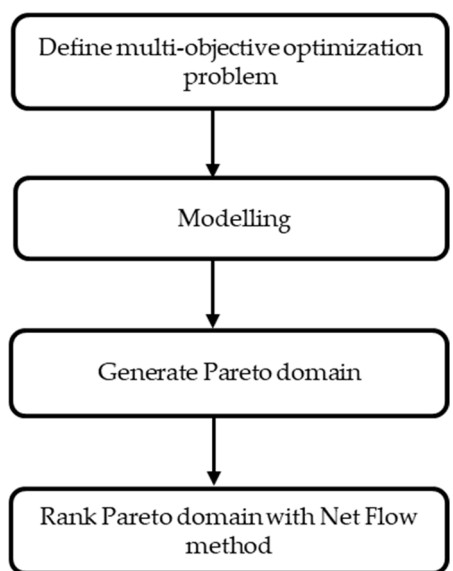

**Figure 5.** Flowchart of the methodology for solving the multi-objective optimization problem.

### 4.1. Definition of the Optimization Problem

In multi-objective optimization, it is desired to determine the best set of operating conditions (decision variables) that will lead to an ensemble of objective functions that is considered optimal, based on the knowledge and expertise of the decision-maker. There is no unique way to define an optimization problem, and it may involve many iterations before reaching an appropriate problem definition [23]. In this investigation, the ensemble consists of three objective functions that are deemed to have an impact on the efficiency and economics of the process: the $SO_2$ conversion ($X$), the $SO_3$ productivity (Pro), and the catalyst weight ($W$), as defined in Equations (12)–(14). This study desired to maximize the first two objective functions, and to minimize the latter one. A summary of the input variables and the objective functions is presented in Table 4. A multi-objective optimization problem can be defined mathematically in general terms using Equation (15), where it is desired to maximize the ensemble of objective functions ($F(x)$) by choosing the set of decision or input variables ($x$) to satisfy some equality and inequality constraints. The values of each input variable must be drawn within their feasible range (see Table 4 for lower and upper bounds). The feasible ranges of the decision variables were chosen large enough to ensure that all Pareto optimal solutions were well within these ranges. In

addition, various configurations of the catalytic packed bed reactor were evaluated and therefore the number of decision variables varied between two and eight. Finally, there were no equality and inequality constraints used [23].

$$
\begin{aligned}
&\text{Max } \boldsymbol{F}(\boldsymbol{x}) = (X(\boldsymbol{x}), \text{Pro}(\boldsymbol{x}), -W(\boldsymbol{x})) \\
&\text{Subject to } h_j(\boldsymbol{x}) = 0 \qquad\qquad j = 1, \cdots, J \\
&\qquad\qquad\quad g_n(\boldsymbol{x}) \geq 0 \qquad\qquad n = 1, \cdots, N \\
&\qquad\qquad\quad (x_i)_{\min} \leq x_i \leq (x_i)_{\max} \qquad i = 1, 2, \cdots, 8
\end{aligned}
\tag{15}
$$

**Table 4.** Input and output process variables.

| Variables | | |
|---|---|---|
| **Decision Variables** | **Identifier** | **Range and Units** |
| Temperature | | |
| First bed | $T_1$ | 600–900 (K) |
| Second bed | $T_2$ | 600–900 (K) |
| Third bed | $T_3$ | 600–900 (K) |
| Fourth bed | $T_4$ | 600–900 (K) |
| Length | | |
| First bed | $L_1$ | 0.02–2 (m) |
| Second bed | $L_2$ | 0.02–2 (m) |
| Third bed | $L_3$ | 0.02–2 (m) |
| Fourth bed | $L_4$ | 0.02–2 (m) |
| **Objective Functions** | **Identifier** | **Desired Attributes** |
| Conversion (%) | $X$ | Maximize |
| Productivity (mol/kg$_{cat}$ s) | *Pro* | Maximize |
| Catalyst weight (kg) | $W$ | Minimize |

*4.2. Pareto Domain*

The Pareto domain is defined as the collection of solutions taken from the total solution set that are non-dominated when compared to the other solutions within this set [23]. A solution is said to be dominated by another solution if the values of all optimization objectives of the first solution are worse than those of the second. Moreover, a non-dominated or Pareto-optimal solution is obtained if no other feasible solution dominates it. As engineering problems are usually complex, it is not possible to derive an analytical solution to represent the Pareto domain; instead, it is represented by a large number of solutions obtained using an iterative procedure [24]. The Pareto front in the objective space represents the optimal trade-off among all objective functions. If the objectives were not conflicting, which is rarely the case, the Pareto domain would contain a unique solution that maximizes or minimizes all objectives. In this investigation, the conversion and the productivity are clearly conflicting.

For the optimization of the SO$_2$ oxidation to SO$_3$, many design strategies were explored keeping the same set of objective functions but with different numbers of decision variables. The decision variables are the inlet temperature and the length of each catalytic bed. Figure 6 presents a schematic diagram of the optimization process for a given strategy, with four catalytic beds in series resulting in eight decision variables to be considered.

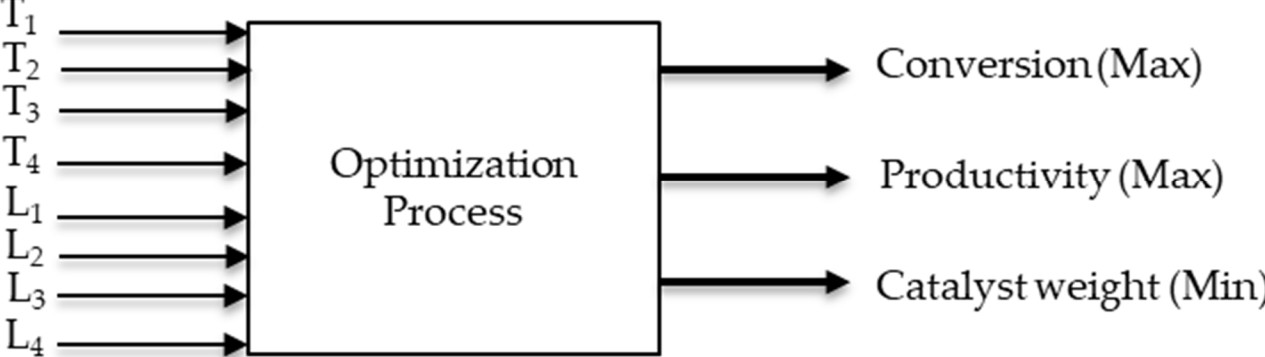

**Figure 6.** Schematic of the optimization of the $SO_2$ oxidation to $SO_3$. This optimization problem considered for the scenario where four catalytic beds in series are used. The eight decision variables are the inlet temperature and the length of each catalytic bed.

### 4.3. Non-Sorting Genetic Algorithm II (NSGA II)

A number of algorithms have been developed to generate a reasonable representation of the Pareto domain from a given number of solutions. One such method, referred to as the Non-Dominated Sorting Genetic Algorithm II (NSGA-II) [22], was used in this investigation to determine a finite representation of the Pareto domain. The evolutionary algorithm NSGA is a popular non-domination based genetic algorithm for multi-objective optimization. It is a very effective algorithm, but has been generally criticized for its computational complexity, lack of elitism, and for choosing the optimal parameter value for sharing parameters [25]. A modified version, NSGA-II was developed, which has a better sorting algorithm, incorporates elitism, and no sharing parameter needs to be chosen a priori. The general procedure of the NSGA-II is summarized in Figure 7.

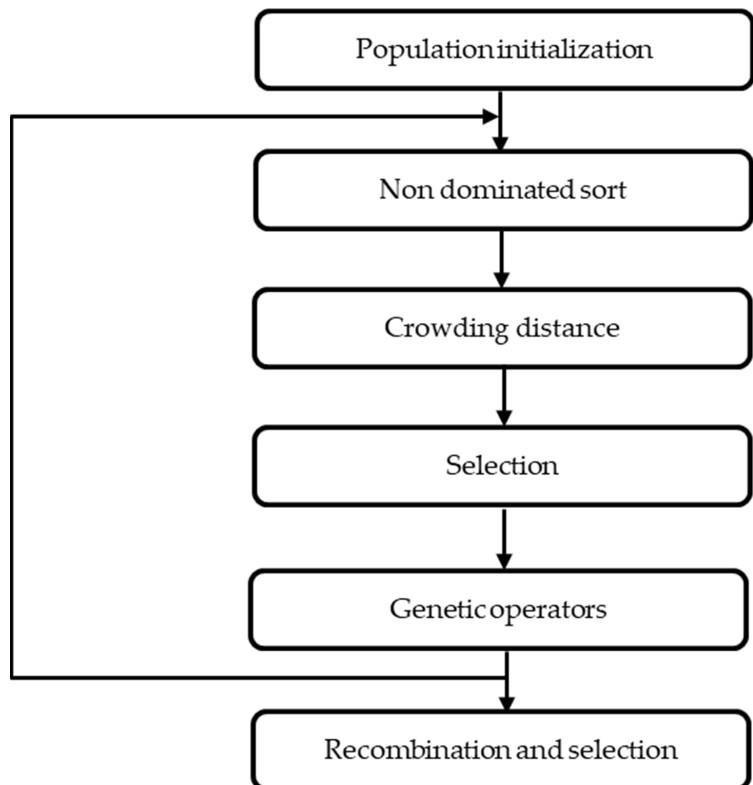

**Figure 7.** Different stages of NSGA-II algorithm are used to adequately circumscribe a large number of representative non-dominated solutions.

The first step in the NSGA-II algorithm consists in generating an initial number of solutions, called the initial population. This is done by randomly selecting values for each of the process inputs or decision variables (inlet temperature and length of each bed) that lie within their lower and upper bounds (see Table 4). The model is then solved for all sets of the input variables within the initial population to generate the three objective criteria associated to each set of decision variables. A pairwise comparison is then performed for all Pareto-optimal solutions in the population to determine the number of times a given solution is dominated, which provides a domination score for each solution. All solutions within the population are then sorted by ascending score of non-domination. The first front is comprised of all solutions with the lowest domination score. The other solutions with higher domination scores are progressively categorized into additional fronts, which indicate their relative performance. A new population of the same size is selected by taking the solutions contained in the fronts with a lower index until the size of the population is exceeded. To make the population equal to the desired size, solutions of the last front are chosen based on the crowding distance to ensure preserving diversity. Parents are selected from the newly created population using binary tournament selection based on the rank and crowding distance. The selected population generates offspring from crossover and mutation operators. The total members of the current population and the current offspring are sorted again based on non-domination, and only the best N individuals are selected, where N is the population size [26]. This procedure is followed until the desired number of generations is performed.

### 4.4. Multicriteria Optimization Algorithm: Net Flow Method (NFM)

When the Pareto domain has been circumscribed, the next step consists of ranking all Pareto-optimal or non-dominated solutions. The Pareto domain has been established through a domination perspective without the bias of a priori knowledge and preferences. However, to rank each solution of the Pareto domain, the decision-maker now needs to express their preferences based on the knowledge that they have of the process. There are many methods for ranking the Pareto domain. In this investigation, a multi-criteria optimization algorithm, known as the net flow method (NFM) [11], has been used. The NFM is essentially an evaluation process, where the decision maker's preferences are included by means of some constraint-like parameters [8]: indifference, preference, veto thresholds, and the relative weights of each objective. A complete description of the NFM and its application to chemical engineering problems are presented in Thibault [27]. The four parameters used for each objective in the NFM are briefly described as follows:

1. The first parameter gives the relative importance of each objective or criterion $k$, expressed as a relative weight ($W_k$). In this algorithm, the weights are normalized:

$$\sum_{k=1}^{3} W_k = 1 \tag{16}$$

2. The second parameter is the indifference threshold ($Q_k$), which defines the range of variation of the difference between the values of the objective k for two solutions for which it is not possible to favour one solution over another for that objective;
3. The third parameter is the preference threshold ($P_k$). When the difference between the values of objective $k$ for two solutions exceeds the preference threshold, the preference is given to solution with the better objective value;
4. The fourth parameter is the veto threshold ($V_k$), which serves to ban a solution relative to another solution as the difference between the values of objective $k$ is too high to be tolerated. The solution with the worst objective is banned based on a particular objective, even if the other criteria are very good. The veto threshold can be equally considered as non-preference information.

The three thresholds are defined for each criterion such that:

$$0 \leq Q_k \leq P_k \leq V_k \tag{17}$$

The values of the four NFM parameters used in this investigation are given in Table 5.

**Table 5.** Relative weights ($W_k$), and indifference ($Q_k$), preference ($P_k$), and veto ($V_k$) thresholds for each objective that are used in the NFM algorithm to rank the entire Pareto domain.

| Criterion (k) | Relative Weight ($W_k$) | Threshold Values | | |
|---|---|---|---|---|
| | | $Q_k$ | $P_k$ | $V_k$ |
| Conversion($X$) | 0.6 | 0.5 | 1 | 3 |
| Productivity(Pro) | 0.3 | 0.002 | 0.004 | 0.01 |
| Catalyst weight($W$) | 0.1 | 500 | 1000 | 4000 |

By defining the ranking parameters (Table 5), the NFM algorithm [8] is implemented to rank the entire Pareto domain based on the decision-maker's preferences. The NFM algorithm is briefly described as follows.

Using a pairwise comparison, the difference $\Delta_k(i,j)$ between of objective k for solutions i and j, the individual concordance index $c_k(i,j)$, the global concordance index $C(i,j)$, and the discordance index $D_k(i,j)$ are calculated using Equations (18)–(21), respectively.

$$\Delta_k(i,j) = F_k(j) - F_k(i) \quad \begin{cases} i \in [1,M] \\ j \in [1,M] \\ k \in [1,K] \end{cases} \tag{18}$$

$$c_k(i,j) = \begin{cases} 1 & \text{if } \Delta_k(i,j) \leq Q_k \\ \frac{P_k - \Delta_k(i,j)}{P_k - Q_k} & \text{if } Q_k < \Delta_k(i,j) \leq P_k \\ 0 & \text{if } \Delta_k(i,j) > P_k \end{cases} \tag{19}$$

$$C(i,j) = \sum_{k=1}^{K} W_k \, c_k(i,j) \begin{cases} i \in [1,M] \\ j \in [1,M] \end{cases} \tag{20}$$

$$D_k(i,j) = \begin{cases} 0 & \text{if } \Delta_k(i,j) \leq P_k \\ \frac{\Delta_k(i,j) - P_k}{V_k - P_k} & \text{if } P_k < \Delta_k(i,j) \leq V_k \\ 1 & \text{if } \Delta_k(i,j) > V_k \end{cases} \tag{21}$$

Using the global concordance and discordance indices, the relative performance of each pair of Pareto-optimal solutions is finally evaluated by calculating each element of the outranking matrix $\sigma(i,j)$ using Equation (22).

$$\sigma(i,j) = c(i,j) \left( \prod_{k=1}^{K} \left[ 1 - (D_k(i,j))^3 \right] \right) \begin{cases} i \in [1,M] \\ j \in [1,M] \end{cases} \tag{22}$$

Finally, the following equation is used to calculate the score of each solution i. The solution with the highest score is the best solution.

$$\sigma_i = \sum_{j=1}^{M} \sigma(i,j) - \sum_{j=1}^{M} \sigma(j,i) \tag{23}$$

The first term in Equation (23) evaluates the extent to which solution *i* performs relative to all the other solutions in the Pareto domain, while the second term evaluates the performance of all the other solutions relative to solution *i* [27]. The solutions are then sorted from highest to lowest according to the ranking score.

Instead of having a unique optimal solution, as in classical optimization methods, the distribution of the numerous Pareto-optimal solutions in the solution space offers the opportunity to better understand the underlying interrelationship that exists between the numerous process variables, and is able to clearly illustrate the trade-offs that are made when examining the different regions of a Pareto domain [28]. The operating conditions associated with the optimal objective function zone are then candidates to be implemented in the process.

## 5. Results and Discussion

To conduct this optimization study as comprehensively as possible, a total of eleven case studies or process scenarios were defined and compared. Table 6 lists these eleven scenarios. The design parameters and operating conditions of the $SO_2$ oxidation packed bed reactor are same for all scenarios and are given in Table 3. The inlet temperature and the length of each catalytic packed bed are under the control of the genetic algorithm, in order to circumscribe the Pareto domain for each process configuration.

**Table 6.** Brief description of the various process scenarios that were optimized in this investigation.

| Number | Scenario |
|:------:|:---------|
| 1 | One catalytic bed |
| 2 | Two catalytic beds with global optimization |
| 3 | Two catalytic beds with individual bed optimization |
| 4 | Three catalytic beds with global optimization |
| 5 | Three catalytic beds with individual bed optimization |
| 6 | Four catalytic beds with global optimization |
| 7 | Four catalytic beds with individual bed optimization |
| 8 | Two catalytic beds, followed by an intermediate absorption column and two catalytic beds with global optimization |
| 9 | Three catalytic beds, followed by an intermediate absorption column and a fourth catalytic bed with global optimization |
| 10 | Four catalytic beds, followed by an intermediate absorption column and a fifth catalytic bed with global optimization |
| 11 | Minimum-length catalytic bed, where the temperature is adjusted to follow the maximum reaction rate curve |

### 5.1. Various Process Configuration Scenarios

The different scenarios of Table 6 differ by the number of catalytic beds used, the presence, or lack of, of an intermediate $SO_3$ absorption column, and the chosen optimization procedure (individual or global) for scenarios with more than one catalytic bed in series. A heat exchanger is used to decrease the temperature of the gas mixture prior to entering the next catalytic bed.

Scenario 1 is the base case, and considers the optimization of only one catalytic bed to determine the optimal solution. Scenarios 2 and 3 consider the optimization of two catalytic beds in series while performing, respectively, a global optimization of the two beds, or optimizing the two beds separately. Similarly, scenarios 4 and 5, as well as 6 and 7, consider the optimization of three or four catalytic beds in series while performing a global optimization of the beds or optimizing the beds separately. With these seven scenarios, it was desired to see the progressive change in the three objective functions as the number of catalytic beds was increased, as well as to observe the gain in performance when the global optimization is performed compared to the bed-by-bed optimization.

In scenarios 8 and 9, four catalytic beds in series are used, with the integration of an absorption column to remove the $SO_3$ produced in earlier beds in order to favor additional

SO$_3$ production in subsequent beds. In Scenario 8, two catalytic beds are used before the intermediate absorption column, and are followed by two catalytic beds, whereas in Scenario 9, three beds are used before, and one after, the absorption column. Similarly, Scenario 10 used an intermediate absorption column after four catalytic beds, and a fifth catalytic bed is included after the intermediate absorption column.

Finally, Scenario 11 differs from the 10 other scenarios as it considers the ideal case where the temperature of a unique catalytic bed at any location along the bed is changed, such that the rate of reaction is always at its maximum for a given conversion instead of using a number of adiabatic beds. In other words, the local temperature of the catalytic bed is adjusted to follow the maximum-rate curve illustrated in Figure 2. This case is considered to provide a point of comparison for the minimum amount of catalysts that would be required to achieve a given conversion under ideal conditions.

### 5.2. Plots of the Decision and Objective Spaces of the Pareto Domain

The Pareto domains for all eleven process scenarios using the evolutionary algorithm NSGA-II were circumscribed with the same ensemble of the three objectives (conversion, productivity, and catalyst weight) and the corresponding decision variables (inlet temperatures of each catalytic bed and the length of each bed) based on the process scenario considered. Each Pareto domain was ranked with the Net Flow Method to determine the optimal region of operation that has given rise to the best Pareto-optimal solutions based on the preferences of a decision-maker.

In this investigation, the eleven Pareto domains had the same shape, even though the Pareto front were located at slightly different locations. As a result, only one Pareto domain is presented and discussed in this paper, and only the highest-ranked Pareto-optimal solution for each scenario is used in the next section to compare the different process strategies.

The results of Scenario 6, presented in Figure 8, are used to illustrate the shape of the Pareto domain. Results are presented in two-dimensional plots, such that the three-dimensional Pareto front is projected onto the two dimensions selected. Results of the ranking by NFM were grouped as follows: the best Pareto-optimal solution, followed by the solutions ranked in the best 5%, the next 45%, and finally the last 50%.

Figure 8a–c presents the graphs of the three objective functions of the ranked Pareto domain. Figure 8a presents the SO$_2$ conversion exiting the fourth catalytic bed as a function of the productivity. This plot clearly shows the typical compromise that commonly exists between conversion and productivity. An increase in conversion is obtained at the expense of a lower productivity, and vice versa. The NFM parameters were chosen in this investigation to favor a higher conversion, as it is required to use as much SO$_2$ as possible and achieve a low concentration at the end of the process, which needs to be captured and disposed of. Therefore, more emphasis was placed on conversion, which is also the case industrially. However, to further increase the conversion beyond the highest-ranked solution, it would significantly decrease the productivity without a significant gain in conversion. Figure 8b presents the plot of the conversion as a function of the total weight of the catalyst for the four beds, whereas Figure 8c presents the plot of the total weight of the catalyst as a function of the SO$_3$ productivity. Figure 8b clearly illustrates that to reach higher conversion, an extremely high amount of catalysts and a much larger total reactor volume would be required. Figure 8c shows that higher productivity would be achieved with a low catalyst weight, leading to a rapid production of SO$_3$, but at a very low conversion. The catalyst weight and the productivity are obviously correlated, as the productivity is defined as the number of moles of SO$_3$ produced per unit time and catalyst weight. It would be possible to use only the conversion and the productivity as objectives, but it was desired to also put a small emphasis on the total quantity of catalyst. If the conversion and productivity would be equally desired, the highest-ranked solution would be located closer to the elbow of the curve in Figure 8b,c.

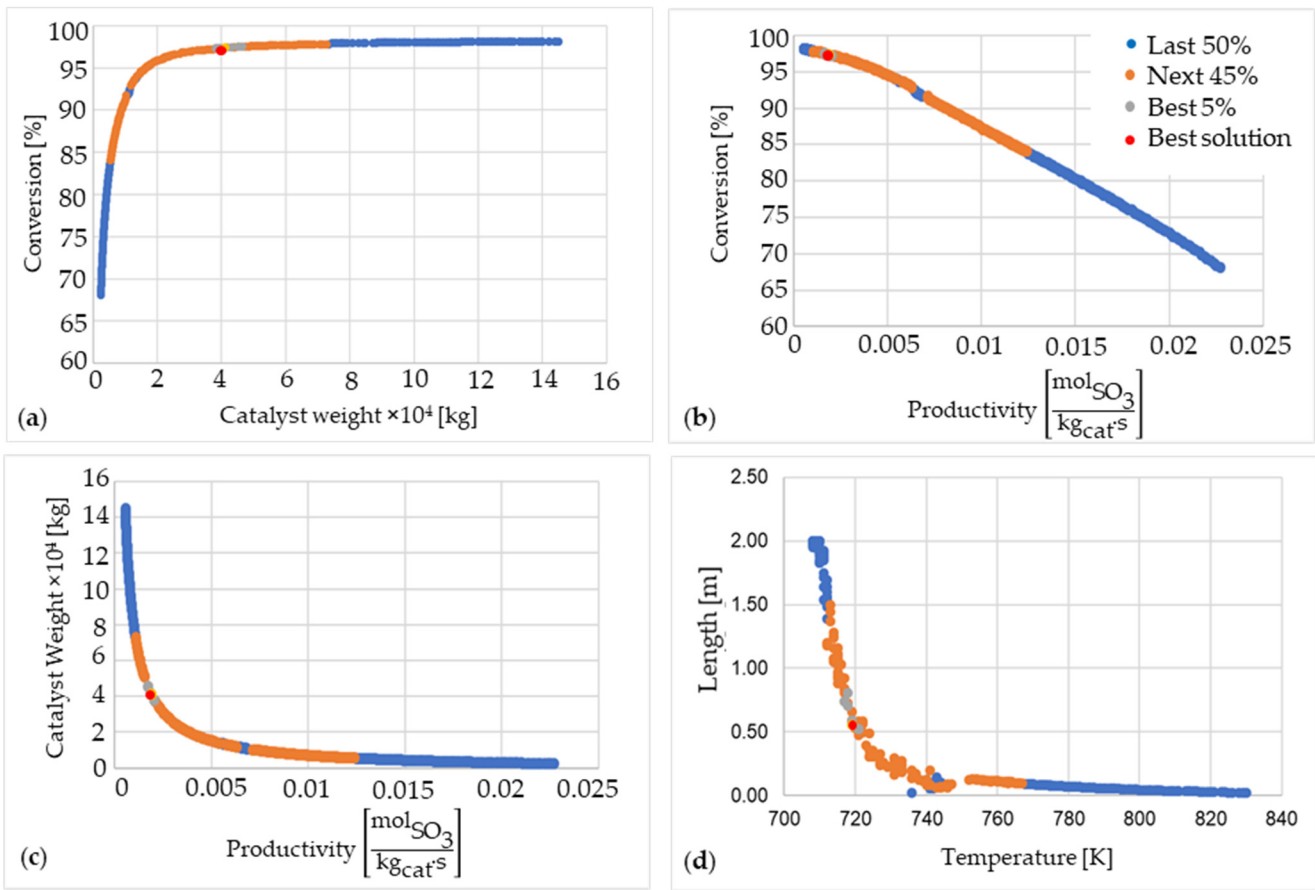

**Figure 8.** Plots of the ranked Pareto domain for the Scenario 6, where four catalytic beds are considered in single contact method. (**a**) conversion versus catalyst weight; (**b**) conversion versus productivity; (**c**) catalyst weight versus productivity; and (**d**) the two decision variables: temperature and length of the last catalytic bed.

For the optimization of Scenario 6, there are eight decision variables, namely the inlet temperature and the length of each catalytic bed. The results of the fourth catalytic bed will be presented here, as all beds show more or less the same pattern between the inlet temperature and the length of the bed. Figure 8d shows the plot of the length of the fourth bed as a function of its inlet temperature. Due to the exothermic equilibrium reaction, if the inlet gas temperature is high, the reaction will reach equilibrium rapidly with a relatively low increase in conversion. On the other hand, a minimum temperature is needed for the catalyst to be active for this oxidation reaction. It is called an auto-ignition temperature, or strike temperature, and is generally in the range of 415 to 425 °C [29]. This higher temperature and the proximity of the equilibrium curve lead to a relatively short length of bed. On the other hand, if the inlet gas temperature is low, the reaction rate will initially be lower and the length of the bed to achieve near-equilibrium will be much longer, whereas the conversion will be much higher, as shown in Figure 2. In addition, the minimum temperature criterion due to the catalyst strike temperature has to be fulfilled for the reaction to occur. The nonlinear relationship observed in Figure 8d is due to the speed of reaction being higher for higher temperatures and the shape of the equilibrium conversion curve.

The ranking of the best 5% of all solutions within the Pareto domain could be affected by the relative weight ($W_k$) of objectives. With a relative weight of unity for one objective and zero for the other two, one could expect that the optimal zone would shift along the Pareto domain towards the extreme of the favored criterion. However, this does not occur to the full extent that would be expected as, although the relative weight for one criterion and its resulting contribution to the concordance index are both zero, the preference and

veto thresholds still play the same role with respect to the discordance index. Therefore, the values of the threshold will affect the ranking of a given solution with respect to another when, in a pairwise comparison, the difference of a given criterion exceeds the preference or the veto thresholds. This sensitivity analysis clearly suggests that the NFM is significantly robust to changes in the relative weights, and the thresholds play a very important role in the ranking of the Pareto domain.

### 5.3. Comparison of Various Process Strategies

Figure 9 shows the highest-ranked Pareto-optimal solutions for the conversion, productivity, and total catalyst weight for scenarios 1 to 10. As can be observed in Figure 9a, the first seven scenarios simply illustrate the progressive and necessary increase in the number of catalytic beds for the production of $SO_3$ to achieve a sufficient conversion, going from 70% to over 97%. The use of four catalytic beds in series has been used industrially for many decades to reach the desired conversion. The results of the four catalytic bed scenarios are then used as a benchmark in this study to investigate the possible improvements implementing other strategies. The target conversion may vary slightly from one industry to another and, to achieve higher conversion given the thermodynamic limitation of this exothermic equilibrium reaction, one would need to sacrifice the productivity and resort to higher amounts of catalyst.

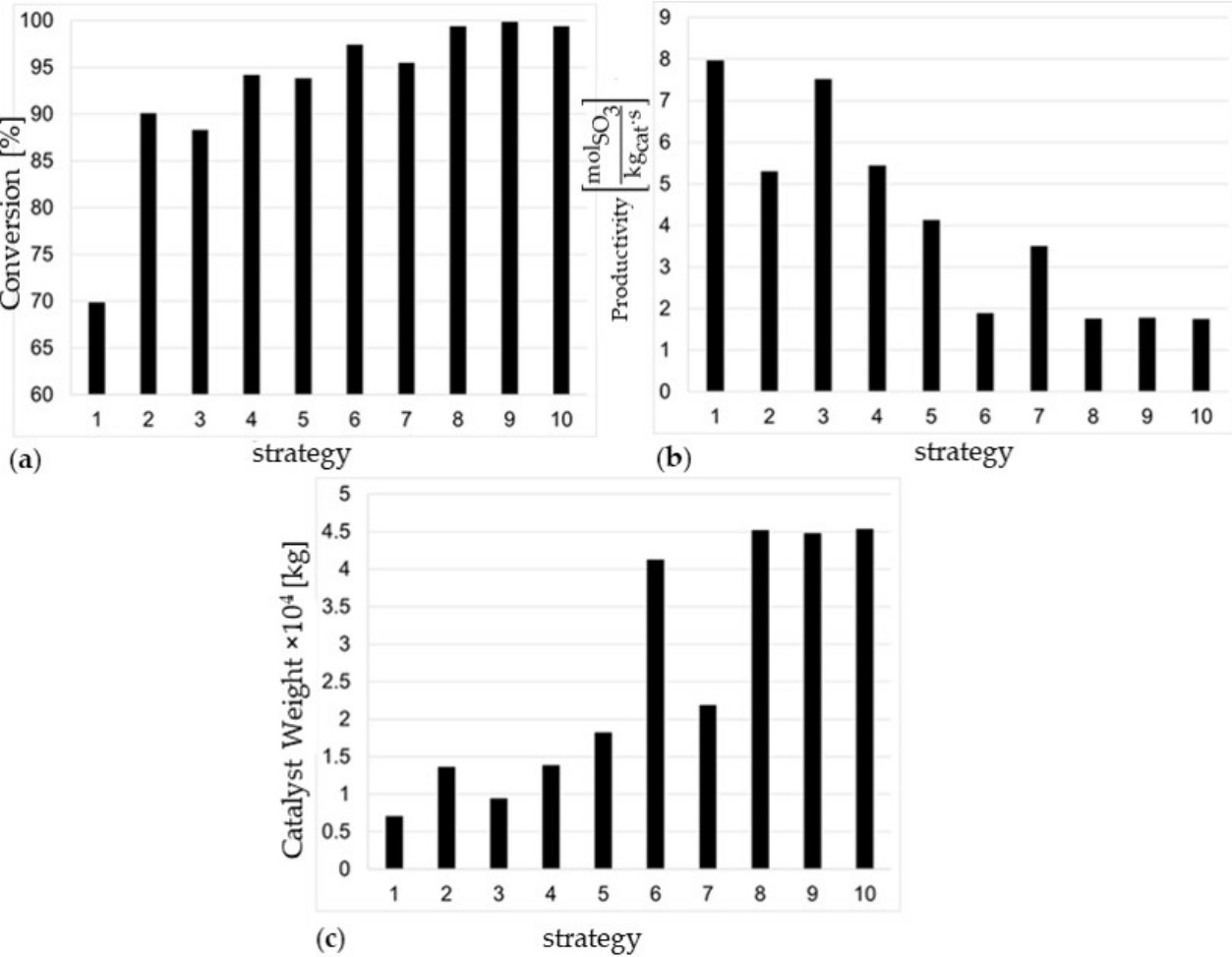

**Figure 9.** Comparison of the optimal values for the three objectives for the various scenarios that are defined in Table 6, based on the number and arrangement of the catalytic packed beds: (**a**) conversion, (**b**) productivity, and (**c**) catalyst weight.

Results of the scenarios 2 to 7 of Figure 9a were also generated to quantify the loss in performance that would occur if the catalytic beds were optimized individually in sequence, rather than being optimized globally with all catalytic beds considered together. As shown in scenarios 3, 5, and 7, for the individual optimization of each bed the conversion is as expected significantly lower in general than for the global optimization, showing the need for such procedure.

Trying to achieve higher conversion by the addition of a fifth catalytic bed did not lead to a significant increase, mainly due to the proximity of the equilibrium curve and the very small driving force, and this scenario was not evaluated further. It is certainly not logical to add a heat exchanger and an extra catalytic bed for a negligible gain in conversion. The double contact process, in which the absorption column is used, is investigated in scenarios 8 to 10 with different process sequences. With these three scenarios, it was possible to achieve a conversion in excess of 99.5% for the highest-rank Pareto-optimal solutions. It is clear that the double contact process permits us to achieve higher conversion and reduce the amount of $SO_2$ that needs to be discarded. The decision to add an intermediate absorption column would be decided from an economical and environmental point of view.

Figure 9b,c present the values of the productivity and catalyst weight for the highest-ranked Pareto-optimal solutions associated to the different scenarios. As mentioned, these two objectives are correlated, as an increase in the catalyst weight leads to a decrease in productivity, as the latter is defined as moles of $SO_3$ produced per second and per kilogram of catalyst. It is clear that to achieve a higher conversion, a much larger weight of catalyst is required, and a much lower productivity needs to be reckoned with. Even if a greater importance has been given to conversion, as it is favored in the industrial production, the ranking algorithm (NFM) attempts to strike a compromise between the three objectives whereby a high conversion is achieved, but without using an excessive amount of catalyst. It is interesting to note that the three scenarios involving the double-contact process (8, 9, and 10) have identical productivity and catalyst weight. This is understandable as the process is limited by the upper limit of conversion, which was nearly reached with these three scenarios.

*5.4. Minimum Length*

In order to determine limiting values of the objectives, the minimum length (or catalyst weight) to reach a conversion equal to the conversion that was obtained with four catalytic beds (Scenario 6) was studied. In this case, the minimum length of a unique catalytic bed would be achieved when the temperature is adjusted at all locations along the packed bed, such that the reaction rate is always at its maximum value. In other words, the conversion-temperature profile needs to follow the maximum-rate curve described in Figure 2. A series of simulations were performed to determine the minimum length of the bed over a range of inlet reactor temperatures (Scenario 11). For a given inlet temperature, the temperature was kept constant until the conversion at which the maximum reaction rate was reached. Thereafter, the temperature was adjusted to follow the constant-rate curve. Results of this study are presented in Figure 10.

Results of Figure 10 show that, for lower inlet temperatures, the length of the variable temperature catalytic bed is relatively high, as a longer portion of the bed is maintained at the inlet temperature where the reaction rate is lower than its optimal value for a given conversion. As the temperature increases, the length of the bed decreases significantly. For comparison, the total length of the four catalytic beds of Scenario 6 has been plotted on Figure 10, which shows that the variable temperature catalytic bed would start to show better performance when the inlet temperature exceeds 790 K. This comparison shows that the length of the four-bed catalytic reactor (Scenario 6) is not significantly different from the best that could be achieved with the large non-adiabatic reactor (about 5% difference in the best case for the same conversion).

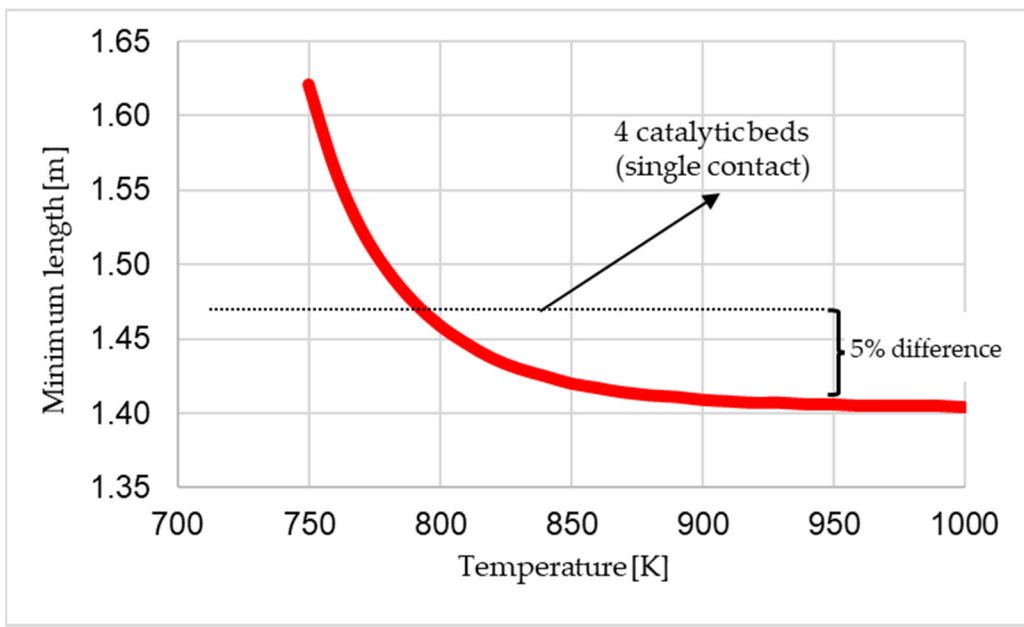

**Figure 10.** Plot of the minimum length of the catalytic bed as a function of inlet temperature (Scenario 11), which is compared with the length of bed of the four catalytic beds for the same optimal conversion.

## 6. Conclusions

The multi-objective optimization of the sulphur dioxide oxidation to the sulphur trioxide was used to determine optimal process scenarios. The optimization of this process considered a series of inputs or decision variables (temperature and length of each bed) and three process objectives ($SO_2$ conversion, $SO_3$ productivity, and catalyst weight). A series of scenarios, or process strategies, were defined and optimized, which aimed in each case to circumscribe the Pareto domain using a genetic algorithm. All Pareto-optimal solutions were then ranked using the Net Flow Method, where the knowledge and preferences of a decision-maker were used. The knowledge of the decision-maker was encapsulated into a relative weight of each objective, along their respective indifference, preference, and veto thresholds.

This analysis allowed the determination of the operating conditions of each scenario, which led to the best trade-off among all three objectives. The highest $SO_2$ conversion was obtained in a double-contact process. Indeed, when the $SO_2$ conversion reaches approximately 90%, it is advantageous to use an intermediate absorption column to recreate favorable conditions for higher $SO_2$ conversion. On the other hand, the four catalytic bed reactor arrangement, commonly used industrially, still provides a very good conversion, which is nearly operating with the minimum length of bed that would be ideally achievable. The next step in this investigation is to perform the global optimization of the sulphuric acid process, which would include the other major process equipment.

**Author Contributions:** Conceptualization, C.F.-L. and J.T.; methodology, M.R.Z., C.F.-L. and J.T.; software, M.R.Z. and J.T.; validation, M.R.Z., C.F.-L. and J.T.; investigation, M.R.Z.; resources, M.R.Z., C.F.-L. and J.T.; data curation, M.R.Z., C.F.-L. and J.T.; writing—original draft preparation, M.R.Z.; writing—review and editing, C.F.-L. and J.T.; visualization, M.R.Z.; supervision, C.F.-L. and J.T.; project administration, C.F.-L. and J.T.; and funding acquisition, C.F.-L. and J.T. All authors have read and agreed to the published version of the manuscript.

**Funding:** Natural Science and Engineering Research Council of Canada (NSERC) is greatly.

**Data Availability Statement:** Additional data can be found in the 2020 MASc thesis of Mohammad Reza Zaker at the University of Ottawa.

**Conflicts of Interest:** The authors declare no conflict of interest.

**Nomenclature**

| | |
|---|---|
| $\varnothing$ | Wilke's Coefficient |
| $\Delta H_R$ | Heat of reaction (J/mol) |
| $\mu$ | Viscosity (Pa·s) |
| $A$ | Reactor area (m$^2$) |
| $C_p$ | Molar heat capacity (J/mol·K) |
| $C_{p,Mix}$ | Molar heat capacity of gas mixture (J/mol K) |
| $D_{bed}$ | Bed diameter (m) |
| $D_p$ | Particle diameter (m) |
| $\varepsilon$ | Void fraction |
| $F$ | Molar flow rate (mol/s) |
| $F_i$ | Molar flow rate of component $i$ (mol/s) |
| $K$ | Number of objectives |
| $k_1$ | Rate coefficient for reaction (Table 1) |
| $K_2$ | Rate coefficient for reaction (Table 1) |
| $K_3$ | Rate coefficient for reaction (Table 1) |
| $K_P$ | Equilibrium constant (atm$^{-0.5}$) |
| $L$ | Length of catalytic bed (m) |
| $M$ | Number of solutions in the Pareto domain |
| $M_w$ | Molecular weight (kg/kmol) |
| $P$ | Total pressure (atm) |
| $p_i$ | Partial pressure (atm) |
| $P_k$ | Preference threshold for criterion $k$ in NFM algorithm |
| $Pro$ | Productivity (mole/kg$_{cat}$ s) |
| $Q_k$ | Indifference threshold for criterion $k$ in NFM algorithm |
| $R$ | Gas constant (8.314 J/mol·K) |
| $r_{e,SO2}$ | Experimental rate for the SO$_2$ oxidation (mol$_{SO2}$/kg$_{cat}$·s) |
| $r_{SO2}$ | Rate of reaction (kmol/kg$_{cat}$·s) |
| $T$ | Temperature (K) |
| $T_{ref}$ | Reference temperature (K) |
| $u$ | Superficial gas velocity (m/s) |
| $V_K$ | Veto threshold for criterion $k$ in NFM algorithm |
| $W$ | Catalyst weight (kg) |
| $W_k$ | Relative weight for each objective criterion |
| $X$ | Conversion (fraction or %) |
| $y_i$ | Mole fraction (fraction or %) |
| $\rho_b$ | Bed density (kg/m$^3$) |
| $\rho_f$ | Fluid gas density (kg/m$^3$) |

**Appendix A. Heat Capacity**

According to Fogler [30], the heat capacity of each species at temperature (T) is frequently expressed as a quadratic function of temperature:

$$C_p = A + B \times T + C \times T^2 \tag{A1}$$

where *A*, *B*, and *C* are coefficients that are given in Table A1.

To approximate the specific heat capacity of a mixture with an infinite number of components when the mole and specific heat capacity of each species are known, the rule of mixture calculator is used:

$$C_{p,\text{ mix}} = \sum_{i=1}^{number\ of\ species} y_i \times C_{p,i} \tag{A2}$$

where $y_i$ is the mole fraction of species *i*.

**Table A1.** Heat capacity coefficients.

| Component | SO$_2$ | SO$_3$ | O$_2$ | N$_2$ |
|---|---|---|---|---|
| $A$ (J/mol·K) | 30.178 | 35.634 | 23.995 | 26.159 |
| $B \times 10^{-3}$ (J/mol·K$^2$) | 42.452 | 71.722 | 17.507 | 6.615 |
| $C \times 10^{-6}$(J/mol·K$^3$) | −18.218 | −31.539 | −6.628 | −2.889 |

**Appendix B. Heat of Reaction**

The heat of reaction of the SO$_2$ oxidation is calculated from the following equation [30].

$$\Delta H_R = \Delta H_R(\text{at the } T_{\text{ref}}) - 6.54(T - T_{\text{ref}}) + \frac{0.0205}{2}\left(T^2 - T_{\text{ref}}^2\right) - \frac{10.007 \times 10^6}{3}\left(T^3 - T_{\text{ref}}^3\right) \tag{A3}$$

The heat of reaction at a reference temperature, 700 K, is −98,787.5 kJ/kmol.

**Appendix C. Gas Viscosity**

Viscosity in gases arises principally from the molecular diffusion that transports momentum between layers of flow [31]. The kinetic theory of gases allows the accurate prediction of the behavior of gaseous viscosity. Within the regime where the theory is applicable, viscosity is independent of pressure and the viscosity increases as the temperature increases. To estimate the dynamic viscosity of an ideal gas as a function of the temperature, Honeywell Unisim was used over the temperature range of 500 < $T$ < 900 (K):

$$\begin{aligned}
\mu_{SO_2} &= 4.67 \times 10^{-8}\, T - 2.23 \times 10^{-6}\ (Pa\ s) \\
\mu_{SO_3} &= 5.68 \times 10^{-8}\, T - 9.09 \times 10^{-6}\ (Pa\ s) \\
\mu_{O_2} &= 4.12 \times 10^{-8}\, T - 5.79 \times 10^{-6}\ (Pa\ s) \\
\mu_{N_2} &= 4.34 \times 10^{-8}\, T - 8.99 \times 10^{-6}\ (Pa\ s)
\end{aligned} \tag{A4}$$

Davidson [32] proposed to use an equation derived from the simple kinetic theory that is easily extended to multi-component systems. It requires the evaluation of a complex coefficient ($\varnothing_{ij}$) for each pair of components in a mixture. The evaluation requires only the viscosity and the molecular weights ($Mw_j$) of individual components:

$$\varnothing_{ij} = \frac{\left[1 + \left(\frac{\mu_i}{\mu_j}\right)^{0.5} \cdot \left(\frac{Mw_j}{M_i}\right)^{0.25}\right]^2}{\frac{4}{\sqrt{2}}\left[1 + \frac{Mw_i}{Mw_j}\right]^{0.5}} \tag{A5}$$

These coefficients are then used with the mole fraction in the calculation of the viscosity of the mixture:

$$\mu_{Mix} = \sum_i \frac{y_i \mu_i}{y_i + \sum_{j \neq i} y_j \varnothing_{ij}} \tag{A6}$$

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
