# Peer review of "Modelling and Multi-Objective Optimization of the Sulphur Dioxide Oxidation Process"

_processes, doi:10.3390/pr9061072_

Round 1
Reviewer 1 Report
The article discloses about optimization of a chemical reactor for a multi-step reaction. The method used for developing the research work is OK. The article is well-writing but some referecences are not appropriately indexed along the article.
Reviewer 2 Report
1) During H2SO4 synthesis, is there a step that produces H2S2O7 (disulfuric acid/ oleum)? Can you add this information to the intro if appropriate?
2) Kinetics of Reaction:
- a) Line 121-129 Can you elaborate more (in a sentence or two) why the second approach was favored over the other? Although the reader can infer why from the paragraph, I think it is good to state it.
3) Line 147 and 150 = where is the reference?
I think you have to check your references for this data/ information.
4) Why did you choose 400 Celsius as the constant temperature for pressure variation studies?
5) Line 198: SO2 – correct the subscript.
6) Figure 9: can you increase the quality/ resolution of this figure?
